# Structural diversity in three-dimensional self-assembly of nanoplatelets by spherical confinement

Da Wang [1,2,7] ✉, Michiel Hermes[1,7], Stan Najmr[3], Nikos Tasios [1], Albert Grau-Carbonell[1], Yang Liu[1,4,6], Sara Bals [2], Marjolein Dijkstra [1], Christopher B. Murray[3,5] & Alfons van Blaaderen [1] ✉

Nanoplatelets offer many possibilities to construct advanced materials due to new properties associated with their (semi)two-dimensional shapes. However, precise control of both positional and orientational order of the nanoplatelets in three dimensions, which is required to achieve emerging and collective properties, is challenging to realize. Here, we combine experiments, advanced electron tomography and computer simulations to explore the structure of supraparticles self-assembled from nanoplatelets in slowly drying emulsion droplets. We demonstrate that the rich phase behaviour of nanoplatelets, and its sensitivity to subtle changes in shape and interaction potential can be used to guide the self-assembly into a wide range of different structures, offering precise control over both orientation and position order of the nanoplatelets. Our research is expected to shed light on the design of hierarchically structured metamaterials with distinct shape- and orientation- dependent properties.

Anisotropic particles such as rods and platelets have been studied for a long time because they show rich phase behaviour including crystal and liquid crystal phases[1]. When the platelets become small enough and their dimension comes into the nanometer range the plates become nanoplatelets (NPLs). NPLs exhibit distinct shape- and orientation- dependent properties due to quantum and dielectric confinement effects. These properties can be used to create novel optoelectronic materials and devices[2–7]. This realisation has lead to an rapid increase in the interest in nanoparticles (NPs) with a plate-like shape over the past decades[2–14]. In order to make optimal use of the enhanced optical properties of the NPLs, it is essential to control the position and the alignment of the NPLs[15,16]. Self-assembly (SA) of the NPLs into superstructures provides a facile platform to control both position and orientation of the NPLs.

Many studies focus on the two dimensional (2D) SA of the NPLs[17–23] and were realised by a so-called liquid-air interface method[24], where a dispersion of the NPLs deposited on an interface between an immiscible liquid subphase and air, forms a film upon drying. In addition to shape diversity of the NPLs, the orientational order of the NPLs in 2D can also be controlled by the kinetics of the evaporation of the solvent[21]. Post-synthetic ligand exchange on the NPLs prior to the 2D SA allows to further control both positional and orientational order of the NPLs, depending on the "softness" and length of the ligands[19,22,23]. However, this method is limited to the formation of (quasi) 2D structures in which the distinct shape- and orientation-dependent properties are different from that of 3D structures. In addition, the superstructures obtained by the liquid-air interface SA method are often transferred onto a substrate, and cannot readily be

[1]Soft Condensed Matter, Debye Institute for Nanomaterials Science, Utrecht University, Princetonplein 5, 3584 CC Utrecht, The Netherlands. [2]Electron Microscopy for Materials Science (EMAT), University of Antwerp, Groenenborgerlaan 171, 2020 Antwerp, Belgium. [3]Department of Chemistry, University of Pennsylvania, Philadelphia, PA 19104, USA. [4]Department of Earth Sciences, Utrecht University, Budapestlaan 4, 3584 CD Utrecht, The Netherlands. [5]Department of Materials Science and Engineering, University of Pennsylvania, Philadelphia, PA 19104, USA. [6]Present address: Monash Centre for Electron Microscopy, Monash University, Clayton, VIC 3800, Australia. [7]These authors contributed equally: Da Wang, Michiel Hermes. ✉e-mail: dawangcolloid@gmail.com; a.vanblaaderen@uu.nl

redispersed into a liquid, limiting their applications. So far, it is still non-trivial to have full control of the orientational order of platelet-shaped building blocks in 3D, hampering the development of new metamaterials with emerging properties. For optimal control over the material properties it is desirable to have a robust path to assemble NPLs into dispersible 3D superstructures with both controllable positional and orientational order.

The SA of NPs in slowly drying emulsion droplets to form so-called supraparticles (SPs, also known as superparticles or clusters) has proven not only to be a powerful method to achieve hierarchical structures in 3D[1,25–29], but also to fabricate functional materials with collective and synergistic properties[29,30]. It should be remarked that in contrast to the 2D superstructures by the liquid-air interface method, SPs obtained from slowly drying emulsion droplets can readily be dispersed in another medium, leading to additional applications[29]. Moreover, an additional advantage of the drying emulsion droplet method is that an additional length scale determined by the droplet size and initial dispersion volume fraction, can be introduced. If all droplets are made to be of the same size, e.g., by shear in a visco-elastic dispersion[31] or by microfluidics[32–36], the resulting SPs can be further used in another SA step, e.g., creating photonic crystals[37].

The SA of spherical NPs in spherical confinement has been studied previously. An unexpected finding, is that up to about 100,000 spheres the SA was influenced by the confinement[38,39]. Instead of a face centered cubic (FCC) crystal phase that is stable in bulk for single component hard spheres (HS), the particles assembled into SPs with an icosahedral symmetry. These more spherically symmetric structures (i.e., icosahedra and surface-reconstructed icosahedra (rhombicosidodecahedra)), formed solely as a consequence of the fact that their entropy was higher in spherical confinement compared to bulk FCC as shown by computer simulations[34,38]. Such icosahedral clusters can also be obtained from the SA of NPs interacting with a Lennard-Jones potential[40]. Intriguingly, when the NPs interact with an attraction of about $2 k_B T$ ($k_B$ and $T$ is the Boltzmann's constant and the absolute temperature, respectively), multi-crystalline FCC domains rather than icosahedra annealed out inside a spherical confinement[41]. Moreover, we elucidated an effect of stoichiometry of binary HS-like particle mixtures on the structures of SPs self-assembled within spherical confinement[42,43]. For rounded cubes we found that icosahedral symmetry was still present, despite the fact that the flat faces of the rounded cubes were still locally orientationally ordered with respect to neighbouring particles[44]. The interplay between spherical confinement, size and number ratio of the NP mixtures, shape and interactions of the NPs, has allowed control over the structure (i.e., stacking, orientation and symmetry) of the SPs[40–44]. This leads to an intriguing question for NPLs: can positional and orientational order of the NPLs be controlled in 3D by using spherical confinement?

In this work, we experimentally demonstrated that disk-, triangular- and leaf-shaped NPLs self-assemble into SPs with different structures including crystals and liquid crystals in slowly drying emulsion droplets, depending on the exact shape of the NPLs. We investigated how spherical confinement affects the stacking of NPLs with these three different shapes in 3D. Computer simulations of the SA of hard disk-shaped platelets in spherical confinement elucidated that both positional and orientational order can be controlled by tuning the aspect ratio as well as the roundness of the NPLs. Icosahedral symmetry emerged in the simulated clusters when the shape of the hard platelets resembled that of a sphere. We obtained both the coordinates and the orientations of the NPLs by state-of-the-art electron tomography, which allowed us to compare our simulation results with our experimental observations on the single particle level in a quantitative manner. Intriguingly, straightly stacked columns observed in the experimental SPs composed of disk-shaped NPLs were absent in the simulated clusters. This indirectly implies that most probably interactions induced by ligands play a role during the experimental SA in spherical confinement, which suppressed the confinement effects. Our work opens up insights to find paths to create metamaterials with controllable positional and orienatational order and extend theoretical studies on the SA of platelet-shaped NPs.

## Results

### Shape of the LnF$_3$ NPLs

In this work, we used lanthanide fluoride (LnF$_3$) NPLs as model systems due to their controllable anisotropic planar shape and recent highlights on their emerging self-assembled superstructures[17–19,22,45,46]. We synthesized 2D LnF$_3$ NPLs by rapid thermal decomposition of lanthanide trifluroacetate precursors in the presence of ligands at high reaction temperatures based on a previous report[18]. The morphology of LnF$_3$ nanocrystals (NCs) can precisely be modulated by the lanthanide source and reaction conditions. Specifically, we focused on three LnF$_3$ (where Ln = Eu, La and Gd in this study) systems with different morphologies: disk-shaped EuF$_3$ (Fig. 1a), triangular LaF$_3$ (Fig. 1b) and leaf-shaped GdF$_3$ NPLs (Fig. 1c). The as-synthesized LnF$_3$ NPLs were subjected to several rounds of purification steps using a solvent/anti-solvent pair combination to minimise the amount of free ligands which may act as depletants during the SA both in bulk and spherical confinement.

### Experimental SA of disk-shaped NPLs in spherical confinement

We first focused on the SA of the disk-shaped EuF$_3$ NPLs capped with oleic acid. The core diameter and thickness of the disk-shaped EuF$_3$

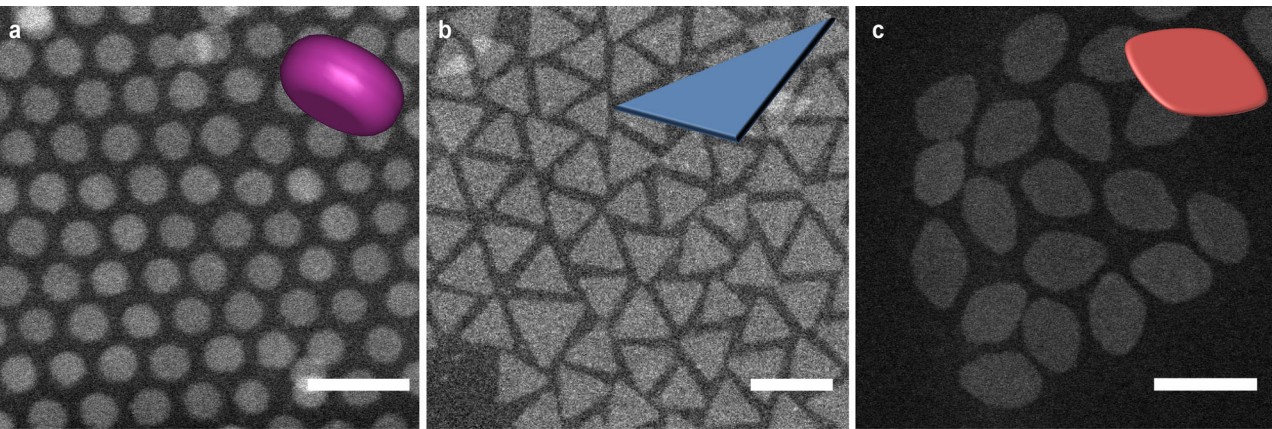

**Fig. 1 | Electron microscopy images and corresponding schematic illustrations of the shape of LnF$_3$ NPLs.** High-angle annular dark-field scanning transmission electron microscopy (HAADF-STEM) images of **a** disk-shaped EuF$_3$, **b** triangular LaF$_3$ and **c** leaf-shaped GdF$_3$ NPLs, respectively. All NPLs were stabilized with oleic acid. Scale bars, 20 nm. Insets, shape illustrations of the NPLs.

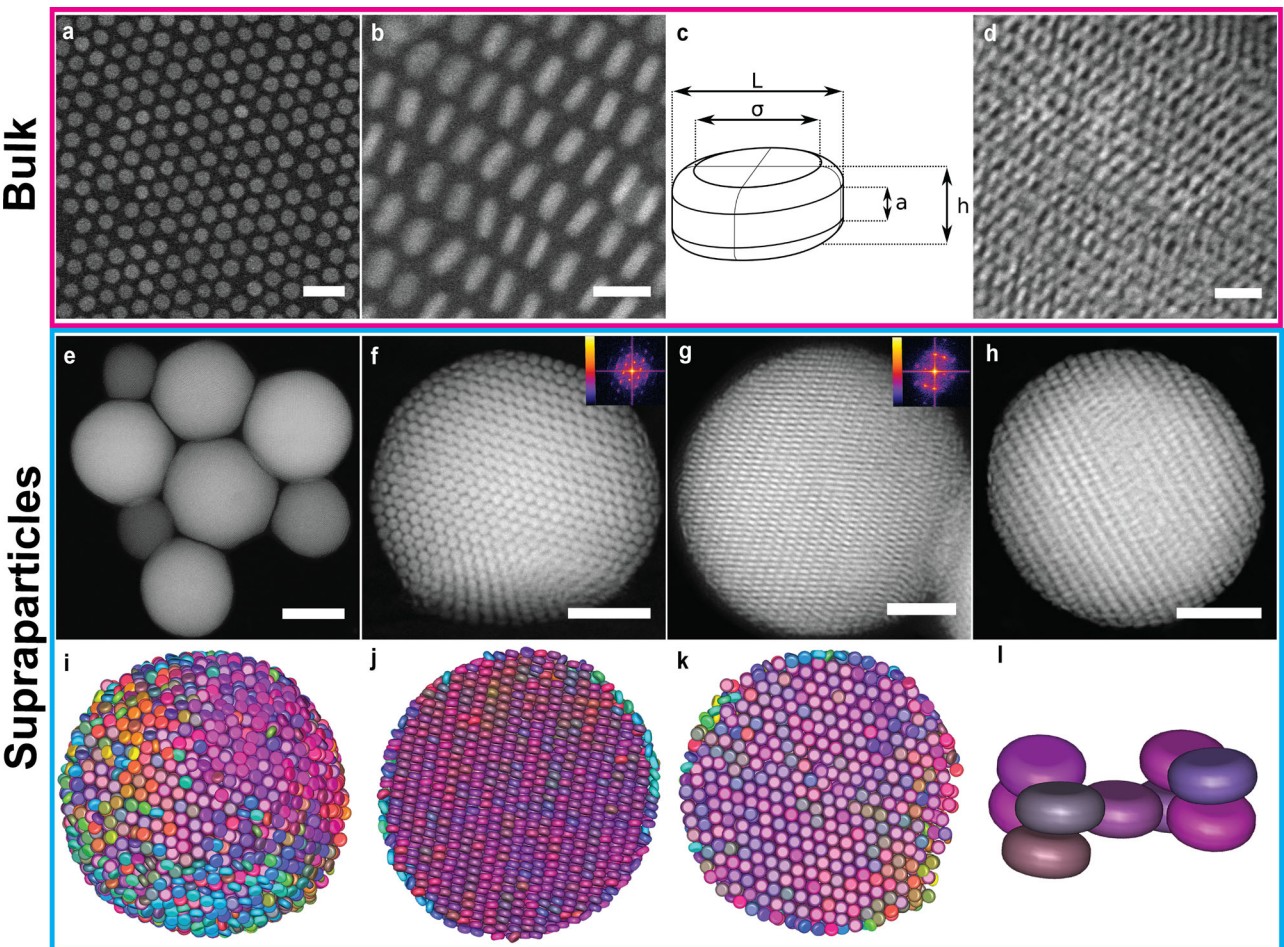

**Fig. 2 | Self-assembled SPs composed of disk-shaped EuF₃ NPLs.** 2D HAADF-STEM images of self-assembled superlattices composed of disk-shaped EuF₃ NPLs showing two types of orientations: **a** A monolayer superlattice lying on the substrate with their flat faces oriented upward and **b** a self-assembled superlattice where the NPLs stack by standing edge-on. **c** A schematic illustration of the disk shape by the Minkowski sum. **d** Self-assembled multi-layer superlattices, showing interdigitation between different columns. **e** Overview of the self-assembled SPs composed of the disk-shaped EuF₃ NPLs. Representative orientations of the SPs. **f** A single SP where the NPLs orient their flat faces towards the confining interface and **g** a SP where the NPLs orient their edges against the confining interface, showing interdigitation between adjacent columns. **h** 2D HAADF-STEM image of a SP with a diameter of 160 nm used for electron tomography and corresponding 3D visualization of the **i** surface termination and **j, k** cross-section structure as viewed **j** perpendicular to and **i, k** along the column stacking direction, respectively. **l** A BCO unit cell extracted from the reconstruction. Scale bars, **a** 20 nm, **b** 10 nm, **d** 20 nm, **e** 200 nm, **f–h** 50 nm. Insets in **f, g**: corresponding fast Fourier transform (FFT) patterns. For an interactive 3D view of the SP, see Supplementary Data 1.

NPLs were 7.8 nm and 2.3 nm, respectively (Figs. 1a and 2a, b). We studied the bulk SA behaviour of the disk-shaped EuF₃ NPLs using a well-developed liquid-air interface assembly technique[24]. The interparticle distance between the two flat faces of neighbouring disk-shaped EuF₃ NPLs was ~ 2.3 nm, which corresponds to interdigitated capping oleats[18,47] (Fig. 2a, b). Taking the ligands into account, the total dimension of the disk-shaped EuF₃ NPLs was 10.2 nm × 4.9 nm, resulting in an aspect ratio of ~ 0.5. We approximately describe the shape of the disk-shaped NPLs by the Minkowski sum of a cylinder with height $a$ and diameter $\sigma$ and a sphere with diameter $d$, resulting in a total height of $h = d + a$ and total diameter $L = \sigma + d$ (Fig. 2c). We define the rounding parameter (or roundness) as $\alpha = a/h$. For $\alpha = 1$ and $\alpha = 0$, the particles are flat cylinders and oblate hard spherocylinders (OHSCs)[48], respectively. Note that the rounding parameter of the experimental disk-shaped NPLs was 0.1. A high-angle annular dark-field scanning transmission electron microscopy (HAADF-STEM) image shows that nearly monodisperse disk-shaped EuF₃ NPLs stacked out of the plane, forming a single layer superlattice with a hexagonal symmetry (Fig. 2a). In other regions of the superlattices, one can also observe ordered lamellar structures, where the disk-shaped EuF₃ NPLs stacked face-to-face and stood on the substrate with their edges perpendicular to the substrate (Fig. 2b). In multilayered domains, adjacent columns of the disk-shaped EuF₃ NPLs interdigitated with each other (Fig. 2d).

To study the SA of the disk-shaped EuF₃ NPLs in spherical confinement, we let the EuF₃ NPLs self-assemble in slowly drying emulsion droplets[38]. All disk-shaped EuF₃ NPLs were self-assembled into highly ordered spherical SPs (Fig. 2e and Supplementary Fig. 1a, b). Moreover, we remark that the five-fold symmetry of an icosahedron which was found in SPs containing HSs[38,39,42] and hard rounded cube[44] systems, was absent in our current study. Instead, all the NPLs formed straightly aligned columns, regardless of the size of the experimental SPs (Supplementary Fig. 1c–e). In some regions, the disk-shaped EuF₃ NPLs were aligned with their flat faces towards the confining interface (Fig. 2f), forming layers viewed along the flat faces of the disk-shaped EuF₃ NPLs. We also observed that some EuF₃ NPLs oriented their edges towards the spherical curvature of the SPs, where the disk-shaped NPLs stacked face-to-face, forming straight columns (Fig. 2g). Each column was interlocked with their adjacent columns, showing a two-fold symmetry (inset of Fig. 2g). We demonstrated the phosphorescence of the isolated disk-shaped EuF₃ NPLs and the self-assembled SPs (Supplementary Fig. 2).

To gain more insight into the self-assembled superstructure, we performed a HAADF-STEM electron tomography study on a SP with a diameter of 160 nm (Fig. 2h; Supplementary Movie 1). We extracted both the positions and orientations of the disk-shaped EuF$_3$ NPLs from the reconstructed tomogram, using an advanced particle tracking technique based on the symmetry of the particle shape[44]. We plot the disk-shaped EuF$_3$ NPLs with colours determined by their orientations (Fig. 2i–k; Supplementary Data 1). In the outermost layer of the self-assembled SP, some EuF$_3$ NPLs oriented their edges against the confining interface whereas some oriented their flat faces towards the interface, forming a defect-rich surface (Fig. 2i). In contrast to the NPLs with different colours at the outermost layer (Fig. 2i), almost all the NPLs have the same colour inside the SP (Fig. 2j, k), indicating that they have the same orientation. We observed that the EuF$_3$ NPLs stacked face-to-face with respect to each other to form straight columns, and every column interdigitated with its neighbouring columns (Fig. 2j). It can be seen that at the two poles of the SP (along the column stacking direction), most of the EuF$_3$ NPLs oriented their flat faces at the outermost layer of the SP (Fig. 2i, j; Supplementary Data 1), maximising face-face interactions of the NPLs. The EuF$_3$ NPLs oriented randomly on the rest parts of the outermost layer (Fig. 2i, k; Supplementary Data 1), which can most probably be ascribed to the deformable cyclohexane-water interface of the emulsion droplets during the SA. Further analysis confirmed that the NPLs inside the assembled SPs formed a body-centered orthorhombic (BCO) crystal (Fig. 2l). In addition, a few twinning defects were found in the self-assembled SP (Supplementary Fig. 3). The structure was further confirmed by a series of 2D projections of a SP with a diameter of 183 nm at different tilting angles and its corresponding 3D representation of electron tomographic reconstruction (Supplementary Figs. 4, 5; Supplementary Movie 2).

## Computer simulations of hard disk-shaped platelets in spherical confinement

To investigate the mechanism that drives the SA, we performed Monte Carlo (MC) computer simulations of disk-shaped platelets interacting through a hard-particle (HP) potential in a slowly shrinking spherical confinement (see Methods section for details). We estimated the van der Waals (vdW) interactions between the inorganic cores of the experimental NPLs, which is close to the thermal energy at room temperature (see Methods section for details; Supplementary Fig. 6). Therefore, we anticipated that attractions between particles played a minor role during the SA.

To identify the effect of the rounding we kept the total height of the particle $h$ and total diameter constant at $h = 0.5L$ and varied the rounding parameter $\alpha$ (Fig. 3). For $0.6 \leqslant \alpha \leqslant 0.9$, most of the disk-shaped platelets showed a single colour in the SPs, indicating a strong orientational correlation across the whole SP (Fig. 3a–d; Supplementary Fig. 7a–d; Supplementary Data 2–5). The SPs had two domes on the pole parts where the director is perpendicular to the spherical boundary (e.g., denoted by two red dashed rectangles in Fig. 3a right column). In the center of the SP, the platelets assembled into a crystal phase, forming a hexagonal arrangement between adjacent columns that can be observed along the column direction (e.g., Fig. 3b right panel). At first glance, it seems that the stacking of the platelets is similar in the simulated clusters, but a more detailed analysis reveals intriguing confinement induced structural differences. Near the confining interface, the disk-shaped platelets lost positional correlation with their neighbouring columns along the direction of the column, forming a hexagonal columnar discotic phase. The loss of correlations allowed the columns to bend to follow the curvature of the spherical boundary. In the two domes, the column shifted in-plane without losing local hexagonal symmetry.

It is expected that hard particles will form the packing with the highest volume fraction, if they do not get stuck in a metastable phase.

For hard disk-shaped platelets there are three competing structures that pack equally well. The first one is the perfect crystal which consists of disk-shaped platelets arranged into columns with a hexagonal arrangement. However, it is possible to shift columns to form a hexagonal columnar phase or to shift planes without changing the density of the optimally packed structure. In the center of the SP the disk-shaped platelets show a bulk-like behaviour while the curvature of the confining interface causes the formation of these two competing phases as they can more easily handle the deformation induced by the hard boundary.

For $\alpha \leqslant 0.5$, more colours are visible, indicating different orientations of the platelets (Fig. 3e–j; Supplementary Fig. 7e–j; Supplementary Data 6–11). For $0.4 \leqslant \alpha \leqslant 0.5$, we observed that the disk-shaped platelets stacked into columns with less marked intracolumnar and intercolumnar correlation in the middle of the SP, resulting in short-range order (Fig. 3e,f; Supplementary Fig. 7e, f; Supplementary Data 6 and 7). We found that the presence of the platelets with random orientations among the columns broke the local hexagonal symmetry (e.g., Fig. 3f right panel). Moreover, we found the boundary between the two poles and the center of the SPs less distinct upon increasing the roundness of the platelets ($\alpha \leqslant 0.3$). Intriguingly, for $\alpha = 0.3$, platelets tended to stack into defect-rich concentric annular layers which propagated from the confining interface towards the center of the clusters (Fig. 3g, h; Supplementary Data 8). The platelets in the middle of the SPs, however, were short-ranged ordered, which probably is due to kinetic effects.

For $0 \leqslant \alpha \leqslant 0.2$, the platelets were randomly packed inside the clusters (Fig. 3i, j; Supplementary Fig. 7g–j; Supplementary Data 9–11). The rounding parameter of the disk-shaped platelets influenced not only the stacking of the platelets inside the SPs but also the stacking in the outermost layer. When the disk-shaped platelets became more rounded ($\alpha \leqslant 0.6$), they tended to orient their flat faces against the spherical confining interface (e.g., Fig. 3e–h; Supplementary Fig. 7e–h). The rounder the platelets were, the more favourable the platelets oriented their flat faces towards the confining interface, thus forming a close-packed hexagonal stacking in the outermost layer of the SP (e.g., Fig. 3g, h). Toroidal rims of the more rounded platelets enabled the ease of different orientations which made the more rounded platelets fill the space more efficiently than sharp-edged platelets[48,49]. It has been reported that the most stable structure of such hard OHSC with an aspect ratio of 0.5 is called an aligned crystal, in which the OHSCs stack into columns with a BCO unit cell[48]. Compared to this aligned crystal, our simulated clusters show a similar structure despite of the absence of the interdigitation between adjacent columns, which can be ascribed to the maximization of the packing fraction at the end of the SA process towards infinite pressures. Although we performed a slow compression in the computer simulations, it is clear that especially for more rounded platelets (e.g., $\alpha = 0$ and 0.1) the final configurations look disordered (Fig. 3i, j; Supplementary Fig. 7i, j). This indicates that the SA of the hard rounded disk-shaped platelets may get kinetically trapped and cannot reach an equilibrium state, which implies that probably a slower compression in computer simulations is required to allow the formation of such aligned crystals.

We further investigated self-assembled structures of the hard disk-shaped platelets by varying the aspect ratio in spherical confinement. For $0.1 \leqslant h/L \leqslant 0.3$, the platelets showed long-range orientational order, forming a defect-rich hexagonal columnar structure where the columns were bent (Fig. 4a; Supplementary Fig. 8a, c and e; Supplementary Data 12–14). The platelets shifted laterally in the dome domains without losing their hexagonal arrangement (Fig. 4b; Supplementary Fig. 8b, d, and f; Supplementary Data 12–14). For $0.4 \leqslant h/L \leqslant 0.5$, the platelets started to loose both positional and orientational order (Fig. 4c, d; Supplementary Fig. 8g; Supplementary Data 15), although the platelets in the outer two layers of the shell were still orientationally correlated. For $0.6 \leqslant h/L \leqslant 0.7$, the platelets also

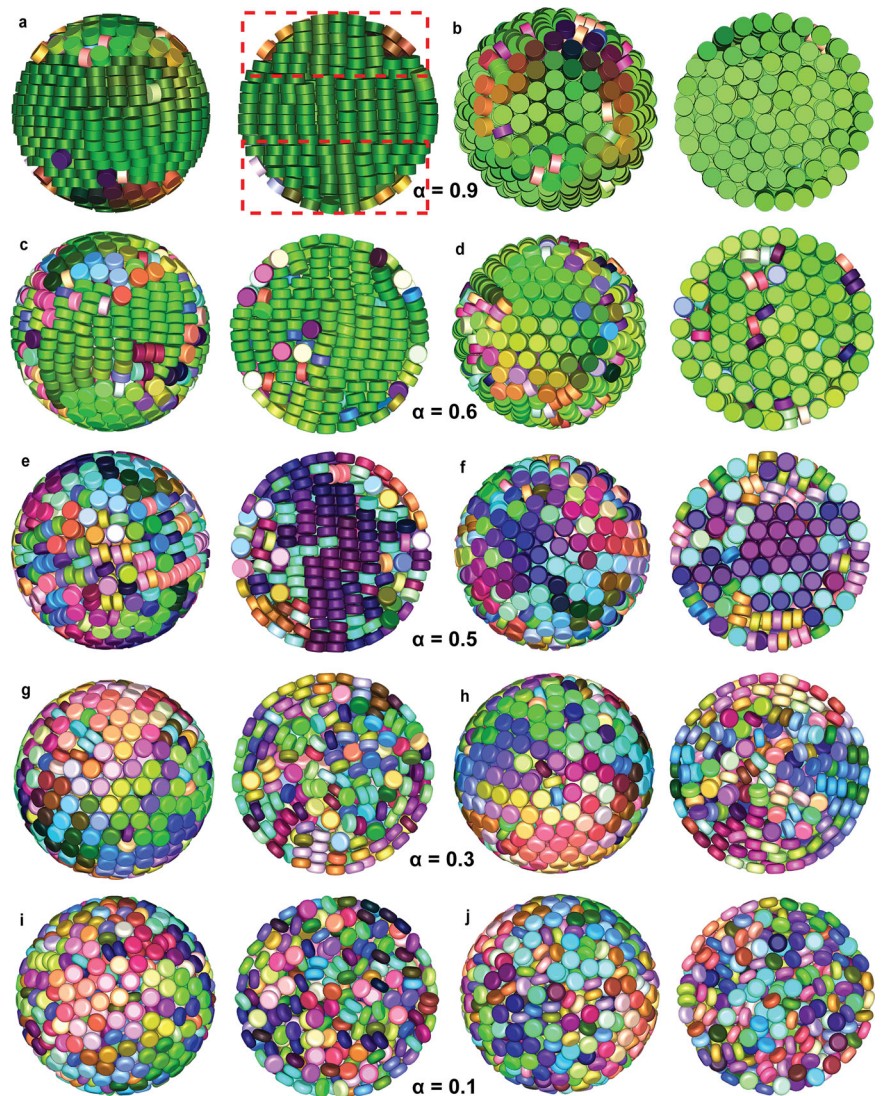

**Fig. 3 | Clusters from computer simulations with varying rounding parameter α.** Fully compressed configurations as obtained from computer simulations with 1000 particles in spherical confinement with different rounding parameters. **a**, **b** $\alpha$ = 0.9, **c**, **d** 0.6, **e**, **f** 0.5, **g**, **h** 0.3, **i**, **j** 0.1, respectively. Note that all simulated clusters are viewed from two directions. The left and right column of each pair, shows a cut-through and a surface view of each simulated cluster, respectively. Different colours represent different particle orientations. Particles are coloured according to their orientations. For all disk-shaped platelets, $h$ = 0.5 $L$. For an interactive 3D view of the clusters, see Supplementary Data 2, 5, 6, 8 and 10, respectively. The dome-structure is highlighted by two red dashed rectangles (right panel of **a**).

lost their orientational order in the outer shells (Supplementary Fig. 8h, i; Supplementary Data 16, 17). Surprisingly, computer simulations showed that an icosahedral symmetry emerged when the aspect ratio further increased to 0.8 (Fig. 4e, f; Supplementary Fig. 8j; Supplementary Data 18–19), resembling the Mackay icosahedral SPs of spheres and rounded cubes interacting with a HP potential[34,38,42,44]. Apparently, a minor change in the shape of hard disk-shaped platelets is sufficient to lead to remarkably different superstructures self-assembled in spherical confinement. It would be intriguing to probe how the rounding parameter of the platelets affects the icosahedral symmetry of the self-assembled superstructure and compare the simulated structures with the experimental ones. However, this research question falls outside the scope of the present work and will be presented in future work.

## Structural disparity between the experimental and simulated clusters

In addition to simulations with 1000 platelets in a slowly shrinking sphere as presented here, we also investigated structures of simulated SPs self-assembled from 500 and 2000 hard platelets (Supplementary Figs. 9, 10; Supplementary Data 20–22). Intriguingly, regardless of the number of particles and therefore the size of the assembled SPs, straightly aligned columns were not observed in any simulated SPs. In contrast to the simulated clusters where the columns were bent, our experimental disk-shaped NPLs formed straight stacked columns inside the SPs and the NPLs intercalated with their neighbouring columns. This subtle but intriguing disparity between the experimental and simulated clusters points to an indirect indication that there are attractive interactions between the experimental NPLs. As we mentioned earlier, the EuF$_3$ inorganic cores interacted through a HP potential (Supplementary Fig. 6). However, we are well aware of the fact that with NP SA it is almost certainly never possible to completely have hard interactions only. For instance, it is well established that for spherical NPs for which the 'hard' core size becomes comparable in size with respect to the length of the stabilizing ligands the colloidal crystal structure that forms is not FCC, associated with HS SA, but instead body centered cubic, which forms at softer repulsive potentials between spherical particles[50,51]. In addition, our experimental NPLs

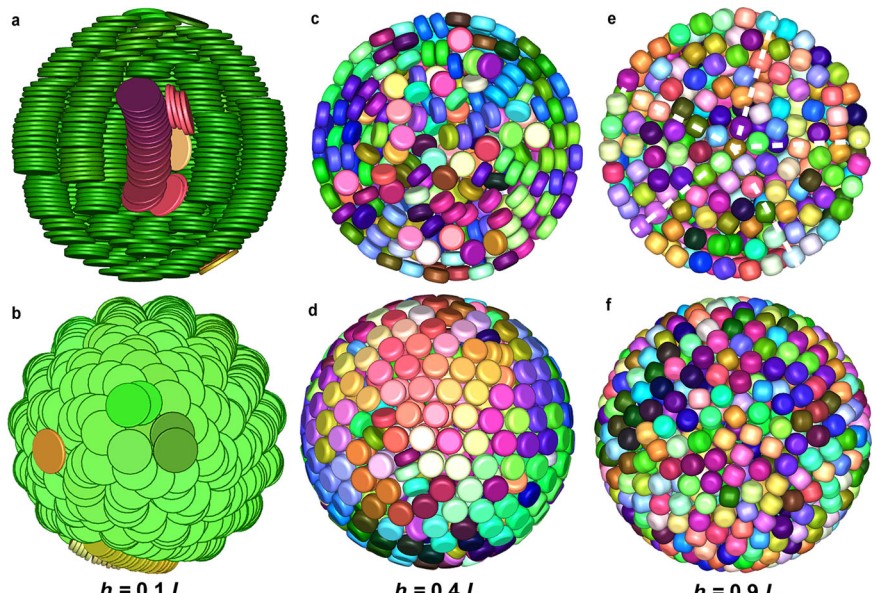

**Fig. 4 | Structural transition from columnar structure to icosahedron.** Fully compressed configurations as obtained from computer simulations with 1000 particles in spherical confinement with different aspect ratio $h/L$. **a**, **b** 0.1, **c**, **d** 0.4, **e**, **f** 0.9, respectively. The top and bottom column of each pair shows a cut-through and a surface view, respectively. The five-fold symmetry of a Mackay icosahedral cluster is readily visible as denoted by the white dashed lines in panel **e**. Note that the viewing angle in panel **b** (i.e., viewed from the pole direction) is perpendicular to that in panel **a**. Particles are coloured according to their orientations. For all particles, $\alpha = 0.3$. For an interactive 3D view of the clusters, see Supplementary Data 12, 15, and 19, respectively. The five-fold symmetry of the icosahedral cluster can be appreciated from the FFT pattern in Supplementary Data 19.

have been purified thoroughly prior to SA, and the amount of free ligands can be considered negligible. We therefore can rule out a substantial role (e.g., depletion interactions) induced by the free ligands during the SA. The attractions most likely result from the interdigitated ligands anchored to the surface of the NPLs[18,19]. This is not surprising as the effective overlap of the ligands on the surfaces (and also the vdW forces) of particles in contact are clearly larger for two flat plates with two parallel faces in contact as compared to that of two spheres, in addition the effective density of the ligands on the curved edges of platelets may be different as well[18,47,52–54]. We therefore believe that the attractions result from the vdW forces between the interdigitated ligands on the flat faces of our experimental NPLs together with the vdW forces generated by the cores of the platelets, which overcome the structural deformation induced by the spherical confinement, thus keeping the columns straightly aligned. Nevertheless, the attractions induced by the ligands rely on the length, conformation and grafting density[19], thus a quantification of attractions between the ligands for our experimental NPLs is not trivial. It would be interesting to investigate in future work how NPLs grafted with different ligands regulate the interactions and determine the structures of the self-assembled SPs.

## SA of triangular NPLs in spherical confinement

In the following part of our study, we shifted our focus to a triangular NPLs system, where anisotropy emerges in the flat basal planes of the LaF$_3$ NPLs stabilized with oleic acid (Fig. 1b). The LaF$_3$ NPLs had an equilateral triangular shape with a side length and thickness of 13.8 nm and of 1.6 nm, respectively (total dimension of 16.1 nm × 4.0 nm including ligands; Fig. 5a, b). The formed defect-rich single-layer superlattice (Fig. 5a) is reminiscent of the honeycomb structure assembled from beveled Au triangular nanoprisms[55]. The geometry of our LaF$_3$ NPLs was close to that of NPLs reported in a previous study[56], which formed a hexagonal close-packed superlattice in bulk. In other regions, the NPLs were found to stack face-to-face, forming ribbonlike arrays (Supplementary Fig. 11a, b). We found that each triangular NPL stood on the substrate and the NPLs were closely packed side-by-side,

where one column of the triangular LaF$_3$ NPLs oriented on the substrate with their vertices and those in adjacent columns stood on their edges (Fig. 5b; Supplementary Fig. 11c). There are two possible ways of stacking such triangular NPLs. The first possible stacking is that the triangular NPLs assemble face-to-face, forming columns. Perpendicular to the direction of the columns, adjacent triangular NPLs form a non-interdigitated yet close-packed structure in the same plane (Supplementary Fig. 12a). The second possible stacking of the triangular NPLs is similar to the first one except that the adjacent columns shift with respect to each other in the direction of the column, forming interlocked superlattices (Supplementary Fig. 12b). Views of slightly tilted NPLs standing either with their vertices or edges in an alternative manner on a substrate (Supplementary Fig. 11b, c) strongly suggest that the self-assembled superlattices correspond to interdigitated ribbons.

The triangular LaF$_3$ NPLs were then assembled into SPs in slowly drying emulsion droplets (Fig. 5c; Supplementary Fig. 13a, b). We observed that some triangular LaF$_3$ NPLs oriented their basal planes towards the spherical confining interface, and therefore formed a triangular lattice with a locally parallelogram lattice (Fig. 5d, e). Electron tomography reconstruction of a LaF$_3$ SP (Fig. 5f; Supplementary Movie 3) clearly showed interdigitation of the NPLs (Fig. 5g). Orthogonal to the basal plane of the LaF$_3$ NPLs, the self-assembled SP exhibited a wallpaper symmetry of p2 (Fig. 5h)[19,57]. We therefore conclude that the triangular LaF$_3$ NPLs assembled into interdigitated arrays with a p2 symmetry, which grew in 3D until meeting the spherical confining interface. We conjecture that the formation of p2 symmetry most probably was induced by lattice gliding during the SA in drying emulsion droplets, which was observed in bulk assembly recently[19]. In addition, dislocations were found in the self-assembled SP, which can be ascribed to a slight size difference of the NPLs during the SA (Supplementary Fig. 14).

## SA of leaf-shaped NPLs in spherical confinement

In the final part, we studied the SA behaviour of leaf-shaped GdF$_3$ NPLs stabilized with oleic acid (Fig. 1c). Superlattices containing leaf-shaped

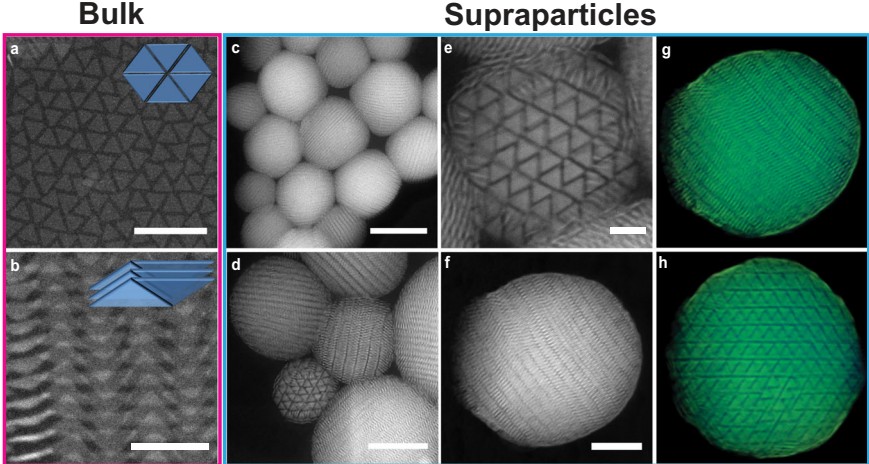

**Fig. 5 | SPs composed of triangular LaF₃ NPLs.** 2D HAADF-STEM images of self-assembled superlattices composed of triangular LaF₃ NPLs **a** with their flat face up, **b** showing an interdigitated ribbonlike array. **c** Overview and **d** a zoom-in view of the self-assembled SPs showing different orientations. **e** An enlarged view of a SP showing a wallpaper p2 symmetry, with the basal planes of the NPLs orienting towards the confining interface. **f** 2D HAADF-STEM image of a SP for electron tomography study. 3D representation of the SP viewed along different directions: **g** The interdigitated ribbonlike arrays and **h** the stacking with a wallpaper p2 symmetry can be visualised (Supplementary Fig. 14). Note that the transparency in **g** and was increased for visual clarity. Insets: illustration of **a** the hexagonal arrangement of a single-layered NPLs and **b** interdigitated ribbonlike arrays. Scale bars, **a** 50 nm, **b** 20 nm, **c** 200 nm, **d** 100 nm, **e**, **f** 50 nm.

GdF₃ NPLs were formed using a liquid-air interface assembly technique[24]. The core dimension of the leaf-shaped GdF₃ NPLs was 15.6 nm and 11.1 nm for the length of the long and short axis, respectively, and 2.3 nm in thickness (Fig. 6a, b). In some regions of the superlattice, the leaf-shaped GdF₃ NPLs laid flat on the substrate and self-assembled into columns out-of-plane, showing a 2D in-plane-short-range positional order and a short-range orientational order (Fig. 6a). The leaf-shape was visible in every single column, which indicates that the NPLs overlap face-to-face with each other. This is a signature of either a columnar or a crystal structure, depending on if long-range-order is present in the third dimension. We also observe that the leaf-shaped GdF₃ NPLs stood on the substrate with their long-axis parallel to the substrate, where the adjacent columns slightly interdigitated (Fig. 6b; Supplementary Fig. 15a). A multi-layered structure showing an ABAB stacking along the direction normal to the substrate, can be visualised in thin film areas (inset in Fig. 6b; Supplementary Fig. 15b). It is however non-trivial to determine if such a stacking is long-ranged. We therefore contend that the leaf-shaped NPLs stacked into a columnar liquid crystal in bulk, which is also backed up by another study[17] despite of a slight shape discrepancy of the NPLs in comparison with our experimental system.

We then applied the slowly drying emulsion droplets methodology to this system. The leaf-shaped GdF₃ NPLs self-assembled into spherical SPs (Fig. 6c; Supplementary Fig. 16). Similarly to the SPs composed of EuF₃ NPLs and LaF₃ NPLs, columns inside the SPs of GdF₃ NPLs interdigitated with their neighbouring columns (Fig. 6d, h; Supplementary Fig. 16). Intriguingly, we found different columnar arrangements with respect to the size of the SPs (Supplementary Fig. 17). The arrangement of the columns inside small SPs (Supplementary Fig. 17c, d) was less ordered compared to that in large SPs (Supplementary Fig. 17a, b). The NPLs aligned their long-axes parallel to the projection plane and stack into straight columns (Supplementary Fig. 17a, b) in large SPs while they showed a short-ranged orientational order in small SPs (Supplementary Fig. 17c, d). To study the structure of the SPs, we performed electron tomography for two SPs with a diameter of 180 nm and 122 nm, respectively (Fig. 6d, h), which we denoted as medium- and small-sized SPs, respectively. Inside the SPs, the leaf-shaped GdF₃ NPLs stacked face-to-face to form columns. Interdigitated columns can be appreciated from the reconstructed orthoslices as well as from the 3D representations (Fig. 6e, f, i, j; Supplementary Movies 4, 5). Surprisingly, in the medium- and small-sized

SPs, rather than forming straight columns where the NPLs oriented almost in the same direction as shown in the large SPs (Supplementary Fig. 17a, b), the leaf-shaped NPLs slightly rotated in-plane (orthogonal to the column direction) with respect to their adjacent NPLs in the same columns (Fig. 6e, i; Supplementary Movies 4, 5). This is reminiscent of the twisted ribbon structure[58,59]. It should be noted that this rotation effect was more pronounced in small SPs (e.g., Fig. 6i), which was most likely induced by the spherical boundary effect of the slowly drying emulsion droplets. Moreover, we found that the columns slightly bent in the medium- and small-sized SPs (Fig. 6e, i; Supplementary Fig. 17c, d). This bending effect was most noticeable for the columns that are closer to the confinement boundary. The NPLs tended to orient their flat faces against the spherical confinement interface, forming a thin shell composed of 2–3 layers of the NPLs (Fig. 6e, i; Supplementary Fig. 17). The self-assembled columns showed a local hexagonal arrangement for the outermost layer (Fig. 6g; Supplementary Fig. 18a, b) as well as the interior part (Fig. 6k) of the SPs viewed along the direction of the columns, which is commensurate with their bulk counterpart (Fig. 6a).

## Discussion

To conclude, we demonstrate a robust strategy to control both the positional and orientational order of NPLs in 3D, exemplified by the SA of three types of lanthanide fluoride NCs with different morphologies into SPs made from slowly drying emulsion droplets. We performed quantitative analysis on the self-assembled SPs composed of NPLs in real space using advanced electron tomography for the first time as far as we know, which was essential to determine not only the positions but also orientations of the disk-shaped EuF₃ NPLs on a single NP level. This most likely enabled us to compare our experimental observations with computer simulated clusters assembled from platelets through a HP potential on a single particle level. The most important difference between the experimental and simulated clusters was that in the simulated clusters, the stacks of the platelets were found to end up being less straight inside the interior of the SPs and close to the interface to be more bent due to geometric frustration by the droplet interface. For the experimental SPs, stacks of the disk-shaped NPLs were equally straight and well ordered. This indicates that there were attractive interactions induced by the ligands between the experimental NPLs, which caused a subtle disparity between the experimental observations and the simulated clusters that were driven by

**Bulk**                                    **Supraparticles**

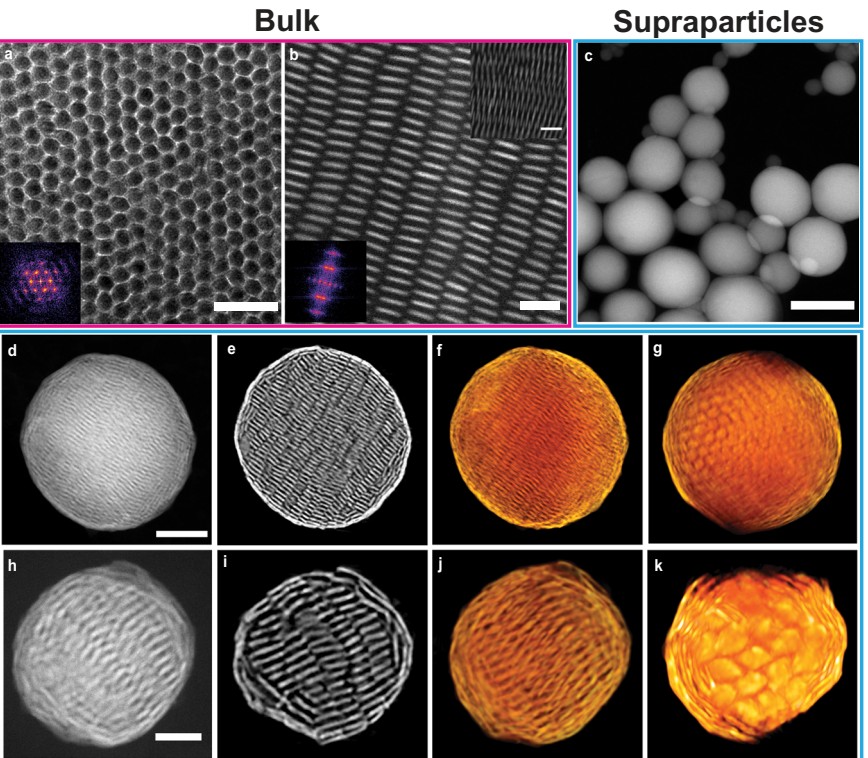

**Fig. 6 | SPs composed of leaf-shaped GdF₃ NPLs.** 2D HAADF-STEM images of self-assembled superlattices composed of leaf-shaped GdF₃ NPLs where the NPLs **a** lay flat face-to-face on the substrate, showing a hexagonal symmetry and **b** align on the substrate with their edges, showing a two-fold symmetry. **c** Overview of self-assembled SPs of leaf-shaped GdF₃ NPLs. **d**, **h** 2D HAADF-STEM images, **e**, **i** Tomographic reconstructed orthoslices and **f**, **j** 3D representations of two self-assembled SPs with a diameter of **d** 180 nm and **h** 122 nm, respectively, showing interdigitation of the NPLs. **g**, **k** Local hexagonal arrangement of the leaf-shaped NPLs along the direction of the columns. Note that the transparency of the 3D representations in **f** and **j** was increased for a better visualisation of the interior structure of the SPs. Insets (lower left) in **a** and **b**, FFT pattern of the whole image. Inset (upper right) in **b**, HAADF-STEM image of a sub-area of the superlattice showing interdigitation of the leaf-shaped GdF₃ NPLs. Scale bars, **a** 50 nm, **b** 20 nm, **c** 500 nm, **d** 50 nm, **h** 20 nm.

entropy alone. In addition, we showed, using computer simulations that hard disk-shaped platelets assembled in a spherical confinement into clusters with an icosahedral symmetry if the shape of the platelets resembled more that of a sphere. Furthermore, we investigated the SA of triangular NPLs and leaf-shaped NPLs into SPs qualitatively, from which we found that the formed SP structure was modulated by the shape of the building blocks.

We envision that our research opens possible future research directions in the field of SA of NPs. First, it would be interesting to probe the strength and range of the interactions between the ligands to see if added interactions can reconcile the structural differences between the simulations and the experiments. Second, systematically tuning softness of the ligands on the NPLs and investigating its role on the self-assembled superstructures (i.e. stacking) from slowly drying emulsion droplets may further expand the library of superstructures self-assembled from NPLs and uncover the interplay between entropy, attractions and spherical confinement, thus shedding more light on the crystallization mechanism in confined space. Third, due to their 4f electron configuration, intriguing optical properties arise from rare-earth elements based materials. Such materials can act as gain media in Whispering Gallery Mode (WGM) microlasers[60]. It will be interesting to investigate for larger SPs if the effects of Mie WGMs can be observed in the emission of these rare-earth based SPs in the future, similarly as was observed for SPs made from semiconductor NPs[61], which can ultimately even lead to lasing of the SPs[36,62] and can potentially be applied in different fields such as biosensing[63]. Finally, in contrast to the bulk assemblies often bound to a substrate that are not trivial to be transferred into liquids, the self-assembled supraparticles by confining building blocks in slowly drying emulsion droplets are

solvent-dispersible colloids. This advantage enables SPs to be dispersible in different liquids and thus they can be employed more easily as building blocks for (nano)device fabrication. We foresee that our experimental methodology that utilises the interplay between shapes, interactions of the building blocks and spherical confinement, is applicable to a broad range of colloidal particles, allowing alternate paths for the design of functional metamaterials with novel properties which rely on positional and orientational order of the building blocks.

## Methods

### Chemicals

Chemicals used were dextran from Leuconostoc mesenteroides (Sigma Aldrich, mol. wt. 1.500.000–2.800.000), cyclohexane (Sigma Aldrich, ≥99.8%), acetone (Sigma Aldrich, ≥99.5%), n-hexane (Sigma Aldrich, ≥99.5%), 1-octadecene (1-ODE, Sigma Aldrich, 90%), sodium dodecyl sulfate (SDS, Sigma Aldrich, ≥99.0%), oleic acid (OA, Sigma Aldrich, ≥99.0%), ethanol (Baker Analyzed, ≥99.9%, absolute), europium oxide (Eu₂O₃, GFS Chemicals, 99.9%), lithium fluoride (LiF, Sigma-Aldrich, ≥99.98%), trifluoroacetic acid (TFA, Fisher Scientific, 99% biochemical grade), potassium nitrate (KNO₃, ≥99.9%, Fischer Scientific), and sodium nitrate (NaNO₃, ≥99.9%, Fischer Scientific), ethylene glycol (EG, Sigma Aldrich, ≥99%). For de-ionized water (DI H₂O) a Millipore Direct-Q UV3 reverse osmosis filter apparatus was used (18 M Ω at 25 °C).

### NC syntheses

Monodisperse LnF₃ NPLs were synthesized by following a previously reported method[18]. Synthesis of RE(TFA)₃: 10 g of RE oxide was added to 100 mL of a 1:1 solution of distilled water and TFA in a round bottom

flask. This suspension was refluxed to 80 °C and stirred until clear. Then, the solution was allowed to cool to room temperature (RT). The solvent was evaporated off, leaving the solid RE(TFA)$_3$ behind.

Synthesis of LnF$_3$ NPLs: RE(TFA)$_3$ (3.6 mmol) and LiF (3.8 mmol) were added into a round-bottom flask with a stir bar and 60 mL of a 1:1 OA/ODE mixture. The flask was connected to a Schlenk line, placed under vacuum, and heated to 125 °C in a silicone oil bath for 45 min. The vessel was then filled with N$_2$ gas and placed in a 1:1 eutectic mixture of KNO$_3$:NaNO$_3$ salt bath. The temperature was raised to 310 °C for EuF$_3$ and GdF$_3$ and 290 °C for LaF$_3$ at a rate of 20 °C/min, respectively. RE choice, reaction conditions, and general batch-to-batch variability provided the different morphologies for the LnF$_3$ NPLs. In this case, the EuF$_3$ and GdF$_3$ was held at temperature for 1 h and the LaF$_3$ was held at temperature for 3 h. The reaction was quenched by adding 15 mL ODE and removing the reaction vessel from the heat source. The NPLs were isolated from the reaction mixture by adding 40 mL of a 1:1 hexane and ethanol solution and centrifuging at $4588 \times g$ for 2 min. The resulting NPLs were dispersed in hexane and no further size-selective precipitation strategies were employed.

## 2D SA of LnF$_3$ NPLs

A $1.5 \times 1.5 \times 1 \text{ cm}^3$ Telfon well and was half-filled with EG. LnF$_3$ NPLs solution ($20 \mu L$) was drop-cast onto the EG surface and the well was then covered by a glass slide to allow slow evaporation of hexane solvent. After 30 min, the formed NPLs film was transferred onto a TEM grid (300 mesh) and was further dried under vacuum in a glovebox to remove extra EG.

## Experimental SA of LnF$_3$ NPLs in spherical confinement

For a typical LnF$_3$ NPLs SA in confinement experiment, 6.5 mg of LnF$_3$ NPLs were dispersed in 1.0 mL of cyclohexane and added to a mixture of 400 mg of dextran and 70 mg of SDS in 10 mL of DI H$_2$O. The resulting emulsion was agitated by shear with a shear rate of $1.56 \times 10^5 \text{ s}^{-1}$, using a Couette rotor-stator device (gap spacing 0.100 mm) following the procedure and equipment described by Mason and Bibette[31]. The emulsion was then evaporated at RT using a VWR VV3 vortex mixer for 48 h. The resulting SPs suspension was purified by centrifugation with a Relative Centrifugal Force of $489 \times g$ for 15 min using an Eppendorf 5415C centrifuge, followed by redispersing in DI H$_2$O. The above-mentioned procedure was repeated twice.

## Electron microscopy sample preparation and measurements

TEM images and electron tomography measurements were obtained with a Thermo Fisher Talos F200X TEM, equipped with a high-brightness field emission gun (X-FEG) and operated at 200 kV.

To prepare a sample for electron tomography analysis, $3 \mu L$ of the SPs suspension in DI H$_2$O were deposited on a Quantifoil (2/2, 200 mesh) copper grid and plunge frozen in liquid ethane using a Vitrobot Mark2 plunge freezer at temperatures around 90 K. The sample was then freeze-dried over a period of 8 h under vacuum at 177 K and subsequently allowed to warm to RT prior to electron microscopy analysis. All tomography studies were performed in HAADF-STEM mode using a Fischione model 2020 single tilt holder. For the 160 nm and 183 nm SPs composed of disk-shaped EuF$_3$ NPLs, the tilt series were obtained within a tilt range from −67° to +71°, from −74° to +68° with an increment of 2°, respectively. For the SP composed of trian-gular LaF$_3$ NPLs, the tilt series was obtained within a tilt range from −72° to +68° with an increment of 2°. For the 180 nm and 122 nm SPs consisting of leaf-shaped GdF$_3$ NPLs, the tilt series were obtained within a tilt range from −72° to +66° and from −64° to +72°, with an increment of 2°, respectively.

The tilt series were aligned using cross-correlation routines implemented in Fiji v1.51p (http://fiji.sc/) and TomoJ (2.31)[64] or using a phase correlation implemented in Matlab[65]. The reconstruction was performed using the Simultaneous Iterative Reconstruction Technique (SIRT)[66] algorithm in TomoJ 2.31 or in the ASTRA Toolbox[67]. 3D visualizations of the reconstructed SPs were implemented in Amira 5.4.0.

## Estimation of an effective pair potential between the disk-shaped EuF$_3$ NPLs

The Hamaker constant of EuF$_3$ in vacuum is not available in the literature. We therefore followed the same approach proposed by Ye et al.[18], approximating the Hamaker constant of EuF$_3$ as that of CaF$_2$. We derived the Hamaker constant of $A_{EuF_3}$ across cyclohexane based on the Hamaker constant of CaF$_2$ in vacuum[68]. The following Hamaker constants in vacuum were used:

$A_{EuF_3} = A_{CaF_2} = 6.88 \times 10^{-20} J$ ($0.43 eV$)[18,69], $A_{cyclohexane} = 5.2 \times 10^{-20} J$ ($0.32 eV$)[68].

The vdW interaction potentials of the experimental NPLs were estimated at experimentally measured inter-platelet distances. Two adjacent platelets with a face-to-face stacking maximizes vdW interactions. The length of fully stretched/extended oleic acid ligands is about 2 nm. The average inter-platelet distance (face-to-face) is 2.27 nm measured by TEM imaging of a dried superlattice, which was subjected to strong drying forces, indicating interdigitation of the ligands. Therefore, the estimated vdW interactions with an inter-platelet distance of 2.27 nm should be considered as upper limits.

The vdW interaction between the plates was modelled as illustrated below[70]:

$$U_{vdW} = \frac{-A}{12 \cdot \pi \cdot \left(\frac{d}{t}\right)^2} \cdot \frac{v}{t^3} \cdot \left(1 + \frac{\left(\frac{d}{t}\right)^2}{\left(2 + \frac{d}{t}\right)^2} - \frac{2 \cdot \left(\frac{d}{t}\right)^2}{\left(1 + \frac{d}{t}\right)^2}\right) \qquad (1)$$

Where $A$ is the Hamaker constant of EuF$_3$ across cyclohexane, $d$ is the inter-platelet distance, $t$ is the plate thickness, and $v$ is the volume of the plate.

At the average experimental particle-particle distance measured from electron microscopy imaging (2.27 nm), the vdW attraction was approximately 0 $k_B$T (Supplementary Fig. 6, dashed line), which was close to the thermal energy at room temperature. It should also be noted that steric repulsive forces given by the ligands when the inter-platelet distance is less than twice of the fully stretched ligand length (4 nm), would further reduce the strength of attractions between the EuF$_3$ cores.

## Optical spectroscopy

The photoluminescence spectra were recorded using an Edinburgh Instruments FLS920 spectrometer, with a 450 W xenon lamp as excitation source and a Hamamatsu R928 photomultiplier tube (PMT) for photon detection.

## Determining the position of the EuF$_3$ SP

The locations and orientations of the individual nanoparticles were determined from the 3D reconstructed tomograms. As a first step we apply a 3D Gaussian blur to the reconstructed data to get rid of the noise. The number of particles $N$ and their rough positions $\tilde{r}_i$ were found by a 3D centroiding algorithm similar to one used by Crocker and Grier[71]. The pixels in the picture were then divided into regions $R_i$ each belonging to a single particle by use of a watershed algorithm. Region $R_i$ is a set of pixels $R_i = \{\{I_1, x_1\}, \{I_2, x_2\}, \ldots \{I_m, x_m\}\}$ with intensity $I_n$ and position $\mathbf{x}_n$ for which water flowing along the steepest ascent would flow to a single local intensity maximum associated with particle $i$. We did not take pixels with an intensity below a threshold of 0.05 (relative to a maximum intensity of 1) into account.

The center of mass, or actually center of intensity, is then measured for all pixels belonging to a single particle to determine the exact

position of the particle

$$\mathbf{r}_i = \frac{1}{I_{\text{tot}}} \sum_{n=0}^{\text{pixels} \in R_i} I_n \mathbf{x}_n, \qquad (2)$$

where $I_{\text{tot}}$ is the sum of all intensities in $R_i$. To obtain accurate coordinates this and all following steps were performed on the unfiltered data. To determine the orientation we used the technique described in the literature[72]. We described the intensity distribution around the center of mass in terms of spherical harmonics. As we know the symmetries of our particles we know that we only need to look at $l = 2$. We can then align the particle by maximizing the correlation between the spherical harmonics of the found particle and the same expansion in spherical harmonics of a reference particle by rotating the expansion using Wigner D matrices. In principle, the intensity can be used directly but we found that for the reconstructed tomography images the gradients are more reliable.

## Computer simulations
We performed Monte Carlo (MC) simulations of 500, 1000, or 2000 hard platelets with height $h$, diameter $L$, and rounding parameter $\alpha$ in the canonical (NVT) ensemble. To model the spherical confinement we used an impenetrable hard spherical wall. To mimic the evaporation of the solvent from the droplets, the diameter of the spherical confinement was slowly reduced at a fixed rate. To avoid structure vitrification and thus the resulting disorder, we initialized the simulations at higher density with all platelets pointing up, similar to a study by Marechal et al.[48] More information can be found in Supplementary Methods Section.

## Data availability
The data generated in this study are provided in the Supplementary Info/Supplementary Data/Source Data files. The data of this study are available from the corresponding authors upon reasonable request. Source data are provided with this paper.

## Code availability
The computer simulations codes used in this study are available upon reasonable request.

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

## Acknowledgements

We thank A. Kadu, M. Chiappini, F. Rabouw, S. Paliwal, X. Xie, C. Xia and Z. Wang for fruitful discussions. D.W. and A.v.B. acknowledge partial financial support from the European Research Council under the European Union's Seventh Framework Programme (FP-2007-2013)/ERC Advanced Grant Agreement 291667 HierarSACol. M.H. was supported by the Netherlands Center for Multiscale Catalytic Energy Conversion (MCEC). D.W. acknowledges an Individual Fellowship funded by the Marie Sklodowska-Curie Actions (MSCA) in Horizon 2020 program (grant 894254 SuprAtom). Y.L. acknowledges the Sustainability project between the faculties of Science and Geosciences of Utrecht University. M.D. acknowledges financial support from European Research Council (Grant No. ERC-2019-ADV-H2020 884902 SoftML). S.B. acknowledges financial support from ERC Consolidator Grant No. 815128 REALNANO. C.B.M. acknowledges support for materials synthesis from the Office of Naval Research Multidisciplinary University Research Initiative Award ONR N00014-18-1-2497. The authors acknowledge the EM square center at Utrecht University for the access to the microscopes.

## Author contributions

D.W. and A.v.B. initiated the experimental part of the project. S.N. carried out nanocrystal syntheses under supervision of C.B.M., D.W. performed self-assembly, electron tomography acquisitions and reconstructions under supervision of A.v.B., M.H. developed advanced particle tracking technique under supervision of A.v.B. M.H. performed computer simulations and data analysis. N.T. performed the initial computer

simulations under supervision of M.D., A.G.C. calculated the vdW interactions of the EuF$_3$ NPLs under supervision of A.v.B., S.B. provided comments on the manuscript and reconstructions. Y.L. assisted with the tomographic reconstruction of a EuF$_3$ SP. D.W., M.H., and A.v.B. co-wrote the paper. All authors analysed and discussed results.

## Competing interests

The authors declare no competing interests.
