## [Peer Review File · Nature Communications]

Structural diversity in three-dimensional self-assembly of nanoplatelets by spherical confinementREVIEWER COMMENTS

Reviewer #1 (Remarks to the Author):

In this paper, the authors have studied the self-assembly of plate-like nanoparticles in spherical confinement by combining experiments and simulations. It is high-quality work and an interesting extension of earlier work on rounded cubes (i.e., Nat. Commun. 9, 2228 (2018)) to rounded disks from the same group. By changing the shape of the nanoplatelets, the authors show a rich phase behavior.

The paper is not well written, which diminishes its value. The introduction is needlessly confusing with mixed tenses. Also, "transition dipoles" are mentioned but never brought up again. The results and discussion have long paragraphs that need to be broken up into individual ideas. Finally, it would strengthen the paper to make more apparent the relevance of their results to physical understanding and/or to engineering. For example, it is not clear why and how the bulk structures and the surface effect they find are advantageous or disadvantageous regarding device synthesis.

Specific points:

- 1) The authors think that the attractive interactions between the experimental NPLs contribute to the disparity between the experimental and simulated clusters. Maybe by reducing the chain length of the ligand chains to increase the repulsive interaction, they could observe similar stacked columns to those found in the simulations.
- 2) Have the authors explored the effects of changing the number of NPs on the overall assembly structures? For example, by increasing the particle number to reduce the confinement effect, they may observe the straight stacked columns observed in the simulations.
- 3) When the aspect ratio increases to 0.8, the platelets with the rounding parameter $\alpha=0.3$ form a Mackay icosahedral cluster with the five-fold symmetry. Did the authors investigate the influence of the rounding parameter value on an icosahedral symmetry for the assembly of NPLs?
- 4) The related simulation software package should be mentioned in the method part.
- 5) The authors mention that the interaction potential can be used to guide the self-assembly of NPLs. However, they do not provide a detailed investigation on how the interaction between NPLs affects the resulting phase structure.

Reviewer #2 (Remarks to the Author):

The manuscript by Da Wang et al. focuses on the self-assembly of platelet-shaped nanoparticles in emulsion droplets. The authors combine experimental work using sophisticated particles with well-defined shapes and excellent uniformity, advanced electron tomographic investigations and simulations to understand the confined self-assembly process. In particular, they show how the spherical confinement influences the self-assembly as a function of platelet shape. I am particularly impressed by the electron

tomography data that provides highly accurate reconstructions of such complex, hierarchical structures. The authors find general agreements between the structures predicted by hard sphere simulations and the experiments, but also provide a detailed discussion on structural discrepancies between experiment and simulation and their origin. Overall, the manuscript is well-written, scientifically convincing and provides an important contribution to confined self-assembly that is of appeal to the broad readership of Nature Communications. I recommend publication of the article with minor revisions and hope that the following comments may be helpful for the authors.

1) In some supraparticles, a deviation from the spherical symmetry of the confinement is obvious, seen very well for example in Supplementary Figure 1. Is that related to the anisotropic particle shape that work against the confinement to improve the resultant structure or caused by other external effects (capillary forces with neighboring droplets....)?

2) Figure 2 in its present form is a bit difficult to read and images of “bulk” crystals prepared from the liquid-interface method (a,b,d), and supraparticles (g,h) seem a bit intermixed when they are shown in the same row. Maybe reordering the figure or adding boxes to spatially separate the types of crystal - or adding cartoon next to it would help to grasp the differences at first glance. The same is true (but less pronounced) for Figs 5 and 6 with the different particle shapes. Here, maybe an inset denoting (“bulk” and “supraparticle”) would be sufficient to clarify the differences.

3) The simulations as a function of roundness of the plate (Fig 3) are intriguing and beautifully show the effect of confinement and the balance between a desire to stack in a columnar way and to align towards the interface. Especially for the cases of intermediate roundness (α between approx. 0.5 and 0.2), this leads to interesting structures such as the annular layers shown of 0.3. For this phase, the authors attribute the short-range order to kinetic effects. With sufficient equilibration time, would the authors expect the platelets in the interior to pack into a columnar fashion or is there simply no way to avoid disorder because of the annular layers templated from the interface?

4) While figures 3 and 4 shows the shape effects on the confined self-assembly very well, an additional, important aspect would be a size (or number) effect. What happens if the number of particles is increased? Will the bulk phases emerge – and if so, at which approximate numbers? Such a numerical experiment, maybe demonstrated for a single shape of intermediate roundness would allow interesting comparisons to the case of spherical particles that the authors have reported in their seminal Nature Materials publication from 2015 (where the threshold is at $\sim 100,000$ particles). Such an effect of confinement size is alluded to in the discussion of the experimental data on leaf structures, where the rotation within a column seems stronger for smaller supraparticles (Fig 6), but simulations may help to provide a more coherent picture of such effects.

5) In the first sentence of the simulation sections, the authors remark that: “Next, we estimated the van der Waals (vdW) interactions between the inorganic cores of the experimental NPLs, which is close to the thermal energy at room temperature (see Methods section for details; Supplementary Fig. 6).” I propose to add a conclusion sentence here (... , therefore, we anticipate that attractions between particles are not important...) that this justifies to use a hard sphere model, which is introduced in the following part.

Reviewer #3 (Remarks to the Author):

1) The authors have demonstrated structural diversity in 3D self-assembly into supraparticles (disk-, triangular- and leaf-shaped nanoplatelets) of nanoplatelets by spherical confinement to understand the confinement effects of nanoplatelets by using three lanthanide (Eu, La and Gd).

2) 2D HAADF-STEM images of self-assembled SPs composed of disk-shaped EuF_3 NPLs, SPs composed of triangular LaF_3 NPLs, SPs composed of leaf-shaped GdF_3 NPLs are conclusive.

3) Experimental findings and the simulated clusters are complementary.

Looking at the novelty and significance of the work can be recommended to be published.

Just one doubt Why below line is incorporated by authors is not conclusive

It will be quite interesting to investigate for larger SPs if the effects of Mie Whispering Gallery modes can be observed in the emission of these rare earth based SPs similarly as were observed for SPs made from semiconductor NPs⁴⁹, which can ultimately even lead to lasing of the SPs

1 Reviewer #1 (Remarks to the Author):

In this paper, the authors have studied the self-assembly of plate-like nanoparticles in spherical confinement by combining experiments and simulations. It is high-quality work and an interesting extension of earlier work on rounded cubes (i.e., Nat. Commun. 9, 2228 (2018)) to rounded disks from the same group. By changing the shape of the nanoplatelets, the authors show a rich phase behavior.

We thank the reviewer for his/her positive comments and his/her appreciation of our work.

The paper is not well written, which diminishes its value. The introduction is needlessly confusing with mixed tenses. Also, “transition dipoles” are mentioned but never brought up again. The results and discussion have long paragraphs that need to be broken up into individual ideas.

We have rewritten large parts the manuscript to be clearer and more consistent and restructured the long paragraphs. We agree with the reviewer that the transition dipoles alignment is not the focus of our current work, we removed it from the revised manuscript.

Finally, it would strengthen the paper to make more apparent the relevance of their results to physical understanding and/or to engineering. For example, it is not clear why and how the bulk structures and the surface effect they find are advantageous or disadvantageous regarding device synthesis.

Following the reviewer’s suggestion, we strengthened the advantages of the supraparticles for device fabrication in the Discussion and conclusion section in the revised manuscript.

Specific points:

1) The authors think that the attractive interactions between the experimental NPLs contribute to the disparity between the experimental and simulated clusters. Maybe by reducing the chain length of the ligand chains to increase the repulsive interaction, they could observe similar stacked columns to those found in the simulations.

We thank the reviewer for his/her suggestions on how the interactions influence the stacking of the experimental NPLs. Indeed, changing the sizes of the ligands on the particles can induce a rich phase behaviour of the particles (*J. Am. Chem. Soc.* **2016**, *138*, 10508–10515; *J. Am. Chem. Soc.* **2021**, *143*, 16163–16172; *J. Am. Chem. Soc.* **2019**, *141*, 1980–1988). Ligands are generally anchored on the NPs to increase the particle stability by the fact that they provide steric stabilization and reduce the van der Waals forces. The van der Waals forces as expressed through Hamaker constants for inorganic materials (*e.g.*, cores of the NPLs in our case) are roughly 10 times larger than between organic materials (*e.g.*, ligands). Therefore, using shorter ligands in general will increase the attractions between the nanoparticles (*Adv. Sci.* **2018**, *5*, 1700179), which would not diminish the attractions between the experimental NPLs. A systematic investigation on how ligand length (and therefore interactions) influences the superstructures assembled from the NPLs will definitely be interesting for a separate follow-up work in the future.

2) Have the authors explored the effects of changing the number of NPs on the overall assembly structures? For example, by increasing the particle number to reduce the confinement effect, they may observe the straight stacked columns observed in the simulations.

We found that the structure of the experimental supraparticles is mostly independent of the number of disk-shaped EuF_3 NPLs. The disk-shaped EuF_3 NPLs formed straightly aligned columns inside the supraparticles regardless of the size of the supraparticles, as can be seen from the revised Supplementary Figure 1 (See Figure 1 of this document).

Figure 1: **Morphology of self-assembled supraparticles (SPs) composed of disk-shaped EuF_3 nanoplatelets (NPLs).** a) Overview, b) a zoomed-in view of the self-assembled SPs. Three SPs with a diameter of c) 100 nm, d) 290 nm and e) 460 nm, respectively, showing that the NPLs stacked into straight columns. The outside NPLs oriented their edges towards the confining interface as can be seen in panels c-d. Scale bars, a) 500 nm, b) 200 nm and c) 20 nm, d) 50 nm, e) 100 nm. Note that a deviation from a spherical geometry in some SPs can be ascribed to capillary forces that are due to the water films in between the SPs in combination with plasticity of the NPL ensemble which was caused by the relatively high volume fraction of the ligands in the SP system.

In the revised main text, we emphasized that the number of disk-shaped NPLs did not affect the structure of experimental SPs, which reads now:

“Moreover, we remark that the five-fold symmetry of an icosahedron which was found in SPs containing HSs ^{38,39,42} and hard rounded cube ⁴⁴ systems, was absent in our current study. Instead, all the NPLs formed straightly aligned columns, regardless of the size of the experimental SPs (Supplementary Figs. 1c-e). ”

We investigated the effect of the number of particles by performing computer simulations with 500 and 2,000 particles as shown in Fig. 2 and Fig. 3 in this document (*i.e.* Supplementary Figs. 9-10). We found that the platelets formed bent columns inside the supraparticles in all cases. Because of kinetic effects, it is non-trivial to reach equilibrium states for larger-sized SPs self-assembled from platelets. Therefore, the number of particles that we can simulate within a reasonable range is, unfortunately, limited. In the revised main text, we added new text in the section of ‘Structural disparity between the experimental and simulated clusters’, which reads now:

“In addition to simulations with 1,000 platelets in a slowly shrinking sphere as presented here, we also investigated structures of simulated SPs self-assembled from 500 and 2,000 hard platelets (Supplementary Fig. 9-10; Supplementary Data 20-22). Intriguingly, regardless of the number of particles and therefore the size of assembled SPs, straightly aligned columns were not observed in any simulated SPs.”

Figure 2: **Clusters obtained from computer simulations of hard disk-shaped particles with varying number of platelets.** Fully compressed configurations as obtained from computer simulations with a,b) 500, c-d) 2,000 platelets with an aspect ratio of $h/L = 0.3$, and e-f) 2,000 platelets with an aspect ratio of $h/L = 0.5$ in spherical confinement. Different colours represent different particle orientations. The left and right column of each pair, shows a surface and cut-through view of each simulated cluster. For all NPLs, $\alpha = 0.5$. For an interactive 3D view, see Supplementary Data 20, 21 and 22, respectively.

Figure 3: **Clusters from computer simulations composed of 2,000 platelets.** Fully compressed configurations as obtained from computer simulations with 2,000 particles in spherical confinement with different shape parameters. a-h): $\alpha = 1.0$ (perfect flat cylinder), 0.8, 0.7, 0.6, 0.4, 0.2, 0.1 and 0 (oblate hard spherocylinders, OHSC), respectively. For all platelets, $h/L = 0.3$. Different colours represent different orientations. The left and right column of each pair, shows surface (left column) and cut-through (right) view, respectively.

3) When the aspect ratio increases to 0.8, the platelets with the rounding parameter $\alpha=0.3$ form a Mackay icosahedral cluster with the five-fold symmetry. Did the authors investigate the influence of the rounding parameter value on an icosahedral symmetry for the assembly of NPLs?

We did not investigate the influence of the rounding parameter on the icosahedral symmetry of the assemblies composed of platelets with an aspect ratio of 0.8.

However, in our recent study (*Nat. Commun.*, **2018**, *9*, 2228), we systematically investigated the influence of the roundness of nanocubes on their self-assembly behaviour in spherical confinement. The flat faces tend to align the anisotropic particles with respect to each other. Sharp corners of the anisotropic particles will push particles away. A small rounding of the corners can drastically change the phase behaviour of the anisotropic particles. Structures of the self-assembled supraparticles depend on the competition between the bulk and spherical curvature-induced stacking, both orientationally and positionally. We believe that such rules are applicable to guide the self-assembly of the platelets as well.

As was shown in the manuscript, when the aspect ratio increases to 0.8, the overall shape of the platelets resembles a rounded cube. We speculate that the flat faces and sharp corners would facilitate the long-ranged ordering of the platelets in spherical confinement at high volume fractions, ending up with aligned columns. In contrast, rounded corners would lose the orientational preference of the platelets, so that they adopt freedom of orientations inside the spherical confinement, leading to the icosahedral symmetry.

It definitely would be interesting to study the effect of the rounding parameters on the icosahedral symmetry as the self-assembly behaviour of such platelets is quite rich. Such a study is beyond the scope of the current work and would fit in a separate research in the future.

We added new text in the main text of the revised manuscript, which reads now:

“It would be intriguing to probe how the rounding parameter of the platelets affects the icosahedral symmetry of the self-assembled superstructure and compare the simulated structures with the experimental ones. However, this research question falls outside the scope of the present work and will be presented in future work”.

4) The related simulation software package should be mentioned in the method part.

The Monte Carlo (MC) code that was used is self-written. The most important part of the code is the overlap algorithm, which is available online (we have added a link, See Supplementary Methods section in the revised Supplementary Information). The visualisation codes that were used are also available online and are now referenced in the Supplementary Method section. We refer the reviewer and readers to the revised Supplementary Information.

5) The authors mention that the interaction potential can be used to guide the self-assembly of NPLs. However, they do not provide a detailed investigation on how the interaction between NPLs affects the resulting phase structure.

We apologise for not have been able to make our points clear.

In the simulated supraparticles, the platelets were interacting with a hard-particle potential. Therefore, the self-assembled superstructures were largely influenced by the curvature of the spherical confinement and exact shapes of the building blocks, forming bent columns in the simulated clusters. In contrast to simulations, possibly slight attractions induced by the ligands were already sufficient to reconcile and counterbalance the structure frustration from the spherical confinement, resulting in the formation of straightly aligned columns in the experimental supraparticles. Therefore, the above-mentioned structure deviation has indicated how the interactions in subtle ways determine the resulting self-assembled superstructures. As elaborated earlier in the answer to question 1, it will be indeed insightful to investigate how the self-assembled superstructures can be tuned by grafting different ligands on the surface of the NPLs then followed by self-assembly as a separate work in the future.

To clarify this point, we added new text in the manuscript, which reads now:

“It would be interesting to investigate in future work how NPLs grafted with different ligands regulate the interactions and determine the structures of the self-assembled SPs.”

2 Reviewer #2 (Remarks to the Author):

The manuscript by Da Wang et al. focuses on the self-assembly of platelet-shaped nanoparticles in emulsion droplets. The authors combine experimental work using sophisticated particles with well-defined shapes and excellent uniformity, advanced electron tomographic investigations and simulations to understand the confined self-assembly process. In particular, they show how the spherical confinement influences the self-assembly as a function of platelet shape. I am particularly impressed by the electron tomography data that provides highly accurate reconstructions of such complex, hierarchical structures. The authors find general agreements between the structures predicted by hard sphere simulations and the experiments, but also provide a detailed discussion on structural discrepancies between experiment and simulation and their origin. Overall, the manuscript is well-written, scientifically convincing and provides an important contribution to confined self-assembly that is of appeal to the broad readership of Nature Communications. I recommend publication of the article with minor revisions and hope that the following comments may be helpful for the authors.

We thank the reviewer for acknowledging the importance of our work and for his/her recommendation of publication in Nature Communications.

1) In some supraparticles, a deviation from the spherical symmetry of the confinement is obvious, seen very well for example in Supplementary Figure 1. Is that related to the anisotropic particle shape that work against the confinement to improve the resultant structure or caused by other external effects (capillary forces with neighboring droplets...)?

The deviation from the spherical geometry in some supraparticles was due to capillary forces induced by drying of the liquid (*i.e.*, water in our case) between the supraparticles when two or more supraparticles were in contact during TEM sample preparation procedure. From Supplementary Fig. 1, one can also see that some supraparticles transformed to concave shaped ones, indicating a plastic deformation. In addition, high volume fraction of

the ligands on the surface of the nanoplatelets are most likely more mobile allowing for freedom of motion of the nanoplatelets, therefore enabling the nanoplatelets to adapt to the deformed geometry. Such a structural deformation can be alleviated by applying freeze-drying (or sublimation) during TEM sample preparation, retaining the original shape of the supraparticles. Therefore, we can rule out the possibility that the supraparticle shape deviation is due to anisotropic particle shape work against the confinement or the deformation of the oil-in-water emulsion droplets. Similarly, de Nijs already investigated the shape deviation of the supraparticles (PhD thesis: Towards Crystals of Crystals of NanoCrystals: A Self-Assembly Study, Dr. Bart de Nijs, <https://web.science.uu.nl/scm/Theses/deNijs.pdf>, Chapter 5, Figure 5.2 and 5.3), which confirmed that the deformation was indeed induced by drying of the water film during TEM sample preparation procedure.

In the revised manuscript, we added new text to explain the reasons for the slightly deformed shape in the caption of Supplementary Fig. 1, which reads now:

Supplementary Figure 1: Morphology of self-assembled supraparticles (SPs) composed of disk-shaped EuF_3 nanoplatelets (NPLs).

a) Overview, b) zoomed-in view of the self-assembled SPs. Three SPs with a diameter of c) 100 nm, d) 290 nm and e) 460 nm, respectively, showing that the NPLs stacked into straight columns. The outside NPLs oriented their edges towards the confining interface as can be seen in panels c-d. Scale bars, a) 500 nm, b) 200 nm and c) 20 nm, d) 50 nm, e) 100 nm. Note that a deviation from a spherical geometry in some SPs can be ascribed to capillary forces that are due to the water films in between the SPs in combination with plasticity of the NPL ensemble which was caused by the relatively high volume fraction of the ligands in the SP system.

2) Figure 2 in its present form is a bit difficult to read and images of “bulk” crystals prepared from the liquid-interface method (a,b,d), and supraparticles (g,h) seem a bit intermixed when they are shown in the same row. Maybe reordering the figure or adding boxes to spatially separate the types of crystal - or adding cartoon next to it would help to grasp the differences at first glance. The same is true (but less pronounced) for Figs 5 and 6 with the different particle shapes. Here, maybe an inset denoting (“bulk” and “supraparticle”) would be sufficient to clarify the differences.

We thank the reviewer’s suggestions to improve the readability. We there-

fore have updated the figures accordingly by adding boxes and denoting “bulk” and “supraparticles” in the revised manuscript.

3) The simulations as a function of roundness of the plate (Fig 3) are intriguing and beautifully show the effect of confinement and the balance between a desire to stack in a columnar way and to align towards the interface. Especially for the cases of intermediate roundness (alpha between approx. 0.5 and 0.2), this leads to interesting structures such as the annular layers shown of 0.3. For this phase, the authors attribute the short-range order to kinetic effects. With sufficient equilibration time, would the authors expect the platelets in the interior to pack into a columnar fashion or is there simply no way to avoid disorder because of the annular layers templated from the interface?

We expect that this is mostly a kinetic effect. Especially for cases where the bulk equilibrium structure is a tilted crystal (*J. Chem. Phys.*, **2011**, *134*, 094501) the nucleation can be very difficult (slow). This tilted crystal structure also does not seem very well matched with the spherical confinement, which is most likely the reason we do not see this structure appear in confinement.

4) While figures 3 and 4 shows the shape effects on the confined self-assembly very well, an additional, important aspect would be a size (or number) effect. What happens if the number of particles is increased? Will the bulk phases emerge – and if so, at which approximate numbers? Such a numerical experiment, maybe demonstrated for a single shape of intermediate roundness would allow interesting comparisons to the case of spherical particles that the authors have reported in their seminal Nature Materials publication from 2015 (where the threshold is at 100.000 particles). Such an effect of confinement size is alluded to in the discussion of the experimental data on leaf structures, where the rotation within a column seems stronger for smaller supraparticles (Fig 6), but simulations may help to provide a more coherent picture of such effects.

We appreciate the reviewer’s suggestions. As was elaborated in the reply to Reviewer 1, we observed the self-assembled superstructures mostly

independent of the size (*i.e.* number of platelets) of the experimental SPs. Straightly stacked columns can clearly be observed in an experimental SP with a size of 460 nm, which shows a strong structural resemblance with respect to assemblies of different sizes. We implemented extra computer simulations with 500 and 2,000 platelets in a spherical confinement and these show very similar behaviour as well. At the moment we can not simulate 100,000 platelets in confinement at a sufficiently slow compression (in comparison with hard spheres, platelets can more easily get kinetically trapped). One would expect that bulk behaviour should have emerged at this point. Somewhere in between we expect a crossover regime, but at the moment we can not be sure where this would be.

5) In the first sentence of the simulation sections, the authors remark that: “Next, we estimated the van der Waals (vdW) interactions between the inorganic cores of the experimental NPLs, which is close to the thermal energy at room temperature (see Methods section for details; Supplementary Fig. 6).” I propose to add a conclusion sentence here (... , therefore, we anticipate that attractions between particles are not important...) that this justifies to use a hard sphere model, which is introduced in the following part.

We thank the reviewer for his/her suggestions.

We therefore added the conclusion sentence in the main text in the revised manuscript, which reads now:

“To investigate the mechanism that drives the SA, we performed Monte Carlo (MC) computer simulations of disk-shaped platelets interacting through a hard-particle (HP) potential in a slowly shrinking spherical confinement (see Methods section for details). We estimated the van der Waals (vdW) interactions between the inorganic cores of the experimental NPLs, which is close to the thermal energy at room temperature (see Methods section for details; Supplementary Fig. 6). Therefore, we anticipate that attractions between particles play a minor role during the SA.”

3 Reviewer #3 (Remarks to the Author):

1) The authors have demonstrated structural diversity in 3D self-assemble into supraparticles (disk-, triangular- and leaf-shaped nanoplatelets) of nanoplatelets by spherical confinement

to understand the confinement effects of nanoplatelets by using three lanthanide (Eu, La and Gd).

2) 2D HAADF-STEM images of self-assembled SPs composed of disk-shaped EuF₃ NPLs, SPs composed of triangular LaF₃ NPLs, SPs composed of leaf-shaped GdF₃ NPLs are conclusive.

3) Experimental findings and the simulated clusters are complementary.

Looking at the novelty and significance of the work can be recommended to be published.

We thank the reviewer for his/her positive remarks and for his/her recommendation for publication in Nature Communications.

Just one doubt Why below line is incorporated by authors is not conclusive It will be quite interesting to investigate for larger SPs if the effects of Mie Whispering Gallery modes can be observed in the emission of these rare earth based SPs similarly as were observed for SPs made from semiconductor NPs⁴⁹, which can ultimately even lead to lasing of the SPs.

We apologise for the confusion caused by the sentence without providing sufficient motivations of explorations of the Mie Whispering Gallery modes in self-assembled SPs.

To improve the readability, we expanded the motivation of exploring Mie Whispering Gallery modes in SPs as an outlook in the Discussion and conclusion section. The updated text reads now:

Third, due to their 4f electron configuration, intriguing optical properties arise from rare-earth elements based materials. Such materials can act as gain media in Whispering Gallery Mode (WGM) microlasers⁶⁰. It will be quite interesting to investigate for larger SPs if the effects of Mie WGMs can be observed in the emission of these rare-earth based SPs similarly as was observed for SPs made from semiconductor NPs⁶¹, which can ultimately even lead to lasing of the SPs^{36,62} and can potentially be applied in different fields such as biosensing⁶³.

REVIEWER COMMENTS

Reviewer #1 (Remarks to the Author):

The authors have addressed all the comments thoroughly and professionally. I recommend the publication of the manuscript without further corrections/additions.

Reviewer #2 (Remarks to the Author):

The authors have addressed all my remarks concerning the initial manuscript. In particular, I follow their argument that increasing the platelet number in simulations is computationally expensive and appreciate the simulations performed with smaller system sizes instead. I believe that in its current form, the paper is well suited to be published in Nature Communications.

1 Reviewer #1 (Remarks to the Author):

The authors have addressed all the comments thoroughly and professionally. I recommend the publication of the manuscript without further corrections/additions.

We thank the reviewer for his/her recommendation of publication in Nature Communications.

2 Reviewer #2 (Remarks to the Author):

The authors have addressed all my remarks concerning the initial manuscript. In particular, I follow their argument that increasing the platelet number in simulations is computationally expensive and appreciate the simulations performed with smaller system sizes instead. I believe that in its current form, the paper is well suited to be published in Nature Communications.

We thank the reviewer for acknowledging our efforts on the current work and for his/her recommendation of publication in Nature Communications.